# The Subjective Experience of Ageism: The Perceived Ageism Questionnaire (PAQ)

**DOI:** 10.3390/ijerph19148792

**Published:** 2022-07-19

**Authors:** Lotte P. Brinkhof, Sanne de Wit, Jaap M. J. Murre, Harm J. Krugers, K. Richard Ridderinkhof

**Affiliations:** 1Department of Psychology, Faculty of Behavioral and Social Sciences, University of Amsterdam, 1018 WS Amsterdam, The Netherlands; s.dewit@uva.nl (S.d.W.); j.m.j.murre@uva.nl (J.M.J.M.); k.r.ridderinkhof@uva.nl (K.R.R.); 2Centre for Urban Mental Health, University of Amsterdam, 1018 WS Amsterdam, The Netherlands; h.krugers@uva.nl; 3Amsterdam Brain & Cognition (ABC), University of Amsterdam, 1018 WS Amsterdam, The Netherlands; 4Faculty of Science, Swammerdam Institute for Life Sciences, University of Amsterdam, 1098 XH Amsterdam, The Netherlands

**Keywords:** aging, ageism, self-perceptions of aging, mental health, quality of life, mental well-being, surveys and questionnaires

## Abstract

Ageism as perceived by older individuals has been recognized as a potential risk factor for physical and mental health. We aimed to develop a comprehensive scale that can quantify perceived ageism among aging individuals (55+), including both positive and negative stereotypes, prejudices, and discriminations. This effort resulted in an 8-item Perceived Ageism Questionnaire (PAQ-8), with good psychometric properties and a two-factor structure distinguishing a positive (3 items) and negative (5 items) subscale (Analysis 1; *n* = 500). This dimensionality was confirmed in a separate cross-validation sample (Analysis 2; *n* = 500). The subscales’ correlation patterns with individuals’ self-perceptions of aging and mental health variables (i.e., quality of life, mental well-being, depression, anxiety, loneliness and perceived stress) accorded with theoretical hypotheses and existing knowledge of the concept of ageism. The PAQ-8 can help to gather more standardized data of the level, role and impact of perceived ageism.

## 1. General Introduction

Although the global rise in life expectancy is one of the greatest achievements of the last century, aging is often portrayed as a gloomy and dreary process, and people hold relatively negative attitudes and perceptions towards older individuals [1]. “Everybody wants to grow old, but nobody wants to *be* old” has been a popular saying since ancient times. Stereotyping, prejudice and discrimination against older adults (even among older adults themselves) on the grounds of their age is commonly referred to as *ageism* (e.g., [2])*,* similar to corresponding ways of casting in sexism and racism. Ageist beliefs and attitudes are thought to emerge from the chronic lifetime exposure to (predominantly negative) societal views and stereotypes about older adults, which are assimilated in and shape people’s feelings, perceptions, and behavior towards older individuals (e.g., [3,4,5]). Ageism can manifest at institutional (e.g., policies), interpersonal (e.g., interactions, such as patronizing older adults), and individual levels (e.g., thinking one is too old to learn something new; [6]). Crucially, ageism is insidious because it is deeply ingrained and more socially accepted than many other forms of bias, and often goes largely unrecognized [6,7,8]. At least one in every two people hold moderately or highly ageist attitudes [9], and more than one in three older people have been a target of ageism [10].

Previous studies have shown that ageism can strongly undermine older adults’ physical and mental health, and argue that it warrants greater recognition as a threat for successful aging [11,12]. Ageism has been associated with earlier death (by up to 7.5 years on average) and slower recovery from disability [13,14]. Furthermore, greater experiences of ageism were found to predict poorer mental health (e.g., depression, anxiety and general stress) and well-being [15,16,17]. Interestingly, older adults who (frequently) *perceive* age discrimination or negative age stereotypes are more likely to have negative self-perceptions of aging [3,17,18], and this negative view of their own aging can lead to decreased purpose in life and increased depressive symptoms [17,18].

Moreover, when negative attitudes about aging are (unconsciously) assimilated (via e.g., stereotype embodiment or stereotype threat; [19,20]), individuals show reduced physical and functional mental health [5,21,22,23], are less likely to engage in preventive health behaviors (e.g., eating a balanced diet, exercising, medication adherence; [24]) and underperform on cognitive or memory tests [25,26,27]. Negative aging attitudes can also lower one’s level of resilience (and thus increase vulnerability) by enhancing one’s negative emotional reactivity to daily stressors [28]. Finally, internalized expectations and stereotypes of loneliness at older age were found to be associated with significantly enhanced loneliness eight years later ([29], i.e., stereotype embodiment [19]). Thus, exposure to age-related prejudices, stereotypes, and discrimination can act as self-fulfilling prophecies (e.g., [30]). Finding ways to combat ageism in the graying population will therefore become increasingly important [31]. 

To further improve our understanding of the effects of ageism, find target points for interventions, and evaluate the efficacy of such strategies to reduce ageism, comprehensive measurements tools are urgently needed [6]. At present, multiple scales are available to examine the *manifestations* (or *prevalence*) of ageism. These scales often reflect stereotypes, prejudices, or expectations towards old age in general (e.g., ‘When people get older, they need to lower their expectations of how healthy they can be’) or towards oneself (e.g., ‘I expect that as I get older, I will get tired more quickly’), and are constructed to evaluate both younger and older adults’ perceptions of aging, aging individuals, and (projected) life as an older individual (see [32] for review). Yet, a validated and comprehensive scale to assess *ageism as perceived and experienced* by older individuals, reflecting all dimensions of ageism, is still lacking. To this end, we aimed to develop and validate a Perceived Ageism Questionnaire (PAQ), that taps into the subjective experiences of older individuals. 

Ageism is considered to include three dimensions: stereotypes, prejudice, and discrimination [33]. Stereotypes are overgeneralized, often simplified ideas that people hold about specific groups of individuals (e.g., ‘Older adults are slow in movement and thought’) and reflect how people think (*cognitive* dimension). Prejudices are opinions related to feelings and attitudes about a person or group of people formed irrespective of evidence or experience (*affective* dimension; e.g., considering older adults as boring conversation partners). Prejudicial reactions are often based on stereotypes. Finally, discrimination is the application of beliefs that are based on stereotypes and prejudices [34] and relates to how people behave towards members of a specific group of individuals (*behavioral* dimension; e.g., older adults are not taken seriously). Most forms of ageism relate to at least two of these dimensions.

Most studies that have looked at *perceived ageism* used self-generated questions that did not always tap into specific age-related situations, but could be applied to several attributes, including age (e.g., gender, race, sexual orientation: ‘How often did people act as if they think you are not smart?’; [17,18]), or did not cover (the full range of) all three dimensions of ageism [17,18,35,36]. Moreover, existing instruments fail to assess the *severity,* or *level*, of perceived ageism [37,38]. A recent study on the influence of ageism on frailty in older adults included three questions to assess perceived ageism that related to all three dimensions (e.g., ‘How often have you felt that someone showed you a lack of respect because of your age, for instance by ignoring or patronizing you?’) and showed high internal consistency [39]. However, this scale did not include any positive forms of ageism. While ageism generally tends to be negative in nature, older people are sometimes also perceived in a positive light [40], such as the stereotype that older adults are wise and know how to respond in complex situations such as social conflicts [7]. Positive ageism may have different, and possibly even opposite, effects on self-perceptions of aging and/or (in)directly on physical, mental and cognitive measures [22,26,41,42,43]. For instance, it was shown that exposing individuals to positive stereotypes of aging can improve gait speed [22], and that positive beliefs about the impact of ageing (e.g., ‘As I get older, I get wiser’) were associated with better mental status, short-delay memory, and executive functioning [41]. Hence, recognizing the distinction between negative and positive ageism is critical, and the development and validation of a comprehensive scale that can assess all dimensions and forms of perceived ageism is of high importance.

In this paper, we describe the results of two sets of analyses (based on existing data from an ongoing, large-scale online study) on the development and validation of the Perceived Ageism Scale (PAQ), including both positive and negative items that relate to all three dimensions of ageism. We conducted two sets of analyses, focused on (1) item selection, scale development, and exploring the reliability and underlying factor structure, and (2) the confirmation of the dimensionality of the PAQ and psychometric as well as validity characteristics in two independent samples. This resulted in an 8-item PAQ. 

## 2. Materials and Methods (Analysis 1)

### 2.1. Participants

For the first analysis, participants were drawn from a larger pool of older individuals of an ongoing online study on successful aging and resilience in the Netherlands [44], consisting of a battery of questionnaires and tests that cover a multitude of relevant factors from multiple domains (e.g., physical, psychological, cognitive, social, environmental; approved by the local ethics committee of the University of Amsterdam, 2020-DP-12556). Participants were eligible to enroll when they were 55 years or older, living in the Netherlands and had no dementia diagnosis. Other exclusion criteria were insufficient command of the Dutch language, impaired vision, or being unable to perform the operations required to successfully use a computer or laptop independently (i.e., mouse clicks, pressing keys on the keyboard). The first 500 participants that completed the full battery of questionnaires and tests were included in the first analysis. This sample size is generally regarded as constituting an adequate basis for identifying factor structures and establishing psychometric properties [45]. The demographic characteristics of the study sample are shown in Table 1.

### 2.2. Construction of the Perceived Ageism Questionnaire (PAQ)

The items of the Perceived Ageism Questionnaire (PAQ) were generated through several steps. Relevant pre-existing measures and the literature were reviewed to formulate multiple candidate items first. A selection of items was made based on their relevance to the construct (based also on informal exchanges with several older adults). A total of 9 items, with 3 items reflecting positive ageism and 6 items reflecting negative ageism, were considered appropriate and relevant to measure the construct of interest. Each of the 9 items described different situations or attitudes that older adults might experience or encounter (see Appendix A for the original, Dutch version of the items, and Table 2 for the translation in English). Participants were asked to report how many times these situations had occurred in the past year using a 5-point Likert Scale (1 = never, 2 = barely, 3 = sometimes, 4 = often, 5 = very often). The items were successfully piloted on formulation and possible ambiguity among an informal panel of 10 participants ranging in age from 55–85. 

### 2.3. Procedure

#### 2.3.1. Item Analysis and Reliability Assessment

Classic item analysis metrics were calculated, including item means and standard deviations, (average) inter-item correlations and corrected item-total scale correlations in R 4.0.0 [47,48] (note that all analyses have been conducted in R unless specified otherwise; item(i) was not included in the total score when calculating the item(i)-total correlation). Average inter-item correlations, as well as the corrected item-total scale correlations, were established for the full PAQ and for the items reflecting either positive or negative ageism separately. The internal consistency (i.e., the proportion of variance attributable to the true score) of the full PAQ and the positive and negative items separately was subsequently assessed using Cronbach’s alpha. Items that showed inadequate metrics were considered to be removed from the scale should it meaningfully (at least 0.01) improve reliability. Principally, alpha becomes lower when the number of items decreases. However, when removing an item improves reliability, this suggests the item may not be adequate.

#### 2.3.2. Dimensionality

The number of common factors needed to account for the correlation among items was assessed using factor analyses. Despite the two anticipated factors reflecting either positive or negative ageism, we recognized that items may also load on non-hypothesized factors (e.g., factors reflecting some specific dimension of ageism; [49]). For that reason, we performed an Exploratory Factor Analysis (EFA) to identify the factor structure of all PAQ items that were retained after the item analysis, and to examine the internal reliability of potential non-hypothesized subscales. The Kaiser-Meyer-Olkin (KMO) test was performed to determine the sampling adequacy for each of the items and for the complete model [50], and Bartlett’s Test of Sphericity was used to establish if the items were correlated enough to be suitable for the EFA [51]. Maximum-likelihood estimation and the oblique rotation algorithm *promax* were adopted, allowing for the factors to be correlated [52]. The number of factors to be retained was established by evaluating the eigenvalues according to the Kaiser Rule, which proposes to only retain factors whose eigenvalues are greater than 1 (i.e., explaining enough variance; [53]). Moreover, items having a factor loading below 0.50 or large cross-loadings (>0.30) were considered for removal from the scale. 

Once the number of factors (and items) was established, a follow-up EFA was performed on the retaining items to obtain the maximum likelihood estimated from the factor loadings and several goodness-of-fit indices were extracted. Indices for good/close fit were set to at least 0.9 for the Tucker Lewis Index (TLI) of factoring reliability [54,55], and less than 0.05 for the Root Mean Square Error of Approximation (RMSEA) and Standardized Root Mean Squared Residual (SRMR; [55,56]). RMSEA values between 0.05 and 0.08 were considered to be ‘acceptable’ [55,57]. Sum scores of the full PAQ as well as all the two anticipated subscales were calculated, and one sample t-tests were performed to determine whether participants experienced the examples at least once (on average). Hence, the sum scores of ‘barely’ (response category 1) were used as criterion value *μ* (mu). 

## 3. Results (Analysis 1)

### 3.1. Item Analysis and Reliability

Descriptive statistics for all items are shown in Table 2. All inter-item correlations are shown in Appendix A. The inter-item correlations of the 6 items reflecting negative ageism (NEG) were all between 0.26 and 0.52, hence only slightly deviating from the range proposed by Clark and Watson [47] (somewhere between 0.15 and 0.50). Moreover, the average inter-item correlations of the negative items were all between 0.32 and 0.45, with *M* = 0.41. The three items reflecting positive ageism (POS) were more strongly related to one another (0.47 to 0.60), with average inter-item correlations ranging from 0.47 to 0.54, with *M* = 0.51. The average inter-item correlation of all items was 0.20. Altogether, this indicates that none of the (sub)scales represented overly broad constructs, but that the items reflecting positive ageism had a relatively narrow focus [47].

The corrected item-total scale correlations among the positive (0.53 to 0.62) and negative (0.43 to 0.63) items were all found to be acceptably high (see Table 2) and above the commonly proposed minimum of 0.30 (e.g., [58,59]). The corrected item-total scale correlations of all nine PAQ items together appeared to be acceptable for the items reflecting negative ageism, but relatively low for those reflecting positive ageism (0.21 to 0.26). Yet, the overall internal consistency of the nine PAQ items was still acceptable (Cronbach’s alpha = 0.67 [0.62–0.71]), and iteratively removing one of the three positive items did not improve the respective Cronbach’s alpha. Hence, no items were removed from the scale. Cronbach’s alphas for the positive and negative items separately were 0.75 [0.71–0.79] and 0.81 [0.78–0.83], respectively. This indicates that positive and negative ageism may indeed reflect two distinctive dimensions of ageism. 

### 3.2. Item Analysis and Reliability

To confirm the distinction between positive and negative ageism, and explore potential other non-hypothesized factors, all nine items were subjected to an EFA. The sampling adequacy sufficed (KMO = 0.80) and Bartlett’s test of sphericity demonstrated that the correlations between items were high enough for the EFA, *Χ²*(8) = 214.9, *p* < 0.001. In line with the item metrics, the EFA reliably identified a simple-structured two-factor model, explaining 46% of the variance, and with all items having a loading on only one of the factors (i.e., no cross-loading; see Table 3). Item 7 (relating to the assumption that older adults cannot hear well) had a factor loading below <0.50 and was therefore removed from the scale (note that the item-total correlation for this item (0.43) was also considerably lower than that for the other negative ageism items (0.56–0.63)). To confirm that this item did not meaningfully contribute to the construct, not even among the oldest subgroup that may experience this situation more often, another EFA that only included individuals of 70 years or older (*n* = 215; the average age for a hearing aid is 70) was conducted. This resulted in similar results, with item 7 having a factor loading <0.50. Table 2 shows the item-total correlations for the remaining 8-item PAQ (henceforth termed PAQ-8, to distinguish it from the original 9-item PAQ pool, henceforth termed PAQ-9).

The EFA on the PAQ-8 identified a two-factor model that explained 49% of the variance (see Table 3 for factor loadings). The first factor had an eigenvalue of 2.30, explained 29% of the variance, and included all five remaining items reflecting negative ageism. The second factor had an eigenvalue of 1.58, explained 20% of the variance, and included all three items reflecting positive ageism. As illustrated in Figure 1, the inter-item correlations of the eight remaining items showed a distribution reflecting a two-dimension scale [60], with positive inter-item correlations among items from the same subscale and negative inter-item correlations among items from the other subscale. Multiple goodness-of-fit indicators were found to be favorable, with a high TLI (0.97), and a relatively low RMSEA (0.051, 90% CI [0.027–0.075]) and SRMR (0.025). These indices were more favorable than observed for the EFA with all nine items (see Table 4). Altogether, these results indicate that, beyond the anticipated distinction between positive and negative ageism, no higher-dimensional factor structure was observed.

### 3.3. Final Scale Statistics

Cronbach’s alpha for the PAQ-8 was 0.63 [0.58–0.68], 0.75 [0.71–0.79], and 0.81 [0.78–0.83] for the full scale, and the POS and NEG subscales, respectively. The PAQ-8 was found to have a normal distribution (*M* = 17.25, *median* = 17, *SD* = 3.48), with scores ranging from 8 to 30 (possible maximum is 40). Scores of the negative (*M* = 7.68, *median* = 7, *SD* = 2.76) and positive (*M* = 9.57, *median* = 10, *SD* = 2.31) ageism subscale ranged between 5 to 23, and 3 to 15, respectively. A one sample *t*-test (against mu) showed that, on average, participants did experience ageism, as reflected in a mean PAQ-8 score that was significantly higher than 16, *t*(499) = 8.03, *p* < 0.001. This was mainly driven by high incidences of positive ageism, *t*(499) = 34.6, *p* < 0.001, rather than negative ageism, *t*(499) = −18.81, *p* = 1. Accordingly, on average, participants perceived more positive than negative forms of ageism, *t*(499) = −37.8, *p* < 0.001. 

## 4. Interim Discussion (Analysis 1)

Our scale development effort resulted in an eight-item PAQ that had good initial psychometric and structural properties, and an adequate level of internal consistency reliability (*α* = 0.63). Most importantly, we found that the reliability was higher when considering the individual subscales of the PAQ-8 (negative: *α* = 0.75, positive: *α* = 0.81), and that the EFA resulted in reliable two-factor structure, distinguishing negative ageism from positive ageism items. This pattern confirms that positive and negative ageism reflect two distinct dimensions of ageism and that they should preferably be examined as separate constructs [40]. On average, individuals in our sample perceived more positive than negative ageism, which paints a relatively favorable picture. However, the considerable individual differences for both positive and negative perceived ageism scores indicate that for many individuals, perceived ageism was substantial, thus underpinning the importance of a scale that can reliably quantify such perceived negative as well as positive ageism.

Based on the EFA, only one item was removed from the initial pool. This item (item 7) aimed to assess the assumption or prejudice that older adults cannot hear well because of their age, but demonstrated no added value to measure perceived (negative) ageism. As the stereotypical assumption that older adults have hearing problems is also one of the reasons that people use ‘elderspeak’ or other forms of patronizing speech [61], this stereotype is perhaps already addressed (at least in part) in item 1 of the scale (which asks how often one feels approached as a child). 

Overall, these results indicate that the PAQ-8 has good initial psychometric properties and that further evaluation and validity testing is appropriate. Our subsequent analyses focus on confirming and extending these findings.

## 5. Introduction (Analysis 2)

The second analysis set out to confirm the factor structure as established and analysed in Analysis 1 in a new, independent sample of participants (*n* = 500) from the same study [44]. Additional purposes were to further evaluate the psychometric characteristics and validity correlates of the PAQ-8 and its NEG and POS subscales. We aimed to explore how PAQ-8 subscale scores vary by age (anticipating that the older the participant, the stronger the perceived ageism), and test theoretically based mediation models to assess whether both positive and negative ageism affect quality of life, mental well-being, depression, anxiety, loneliness, and perceived stress directly, as well as indirectly through their impact on self-perceptions of aging, as assessed by the Aging Perceptions Questionnaire (APQ; comprising seven different dimensions of aging attitudes [62]).

Based on previous studies, we expected to find positive associations with negative ageism for depression [15,17,18,63,64,65], anxiety [15,43], loneliness [29,43,63,65,66], and stress [15,67]. In contrast, negative associations were expected for mental well-being [15,68] and quality of life [69,70]. Previous studies on the mental health benefits of positive ageism are relatively scarce (e.g., [42]: well-being; [43]: anxiety). Yet, considering the positive benefits reported for other domains (e.g., memory function, physical recovery; [22,26,41]) that are known to promote mental functioning, we expected to find rather favourable relationships with mental health, mental well-being and quality of life. As a previous study by Levy and colleagues [67] showed that positive ageism can provide a buffer against cumulative stress at old age by stabilizing cortisol levels, but not lowering stress levels, we anticipated that scores on our positive ageism subscale may not be associated with reduced perceived stress levels. 

We expected that these effects would, at least partially, be mediated by self-perceptions of aging. In line with increasing support for the idea that exposure to age-related stereotypes, prejudices, and discrimination can impact one’s self-perceptions of aging [19,30], we anticipated to find coherent relationships between positive and negative ageism subscale scores and participants’ attitudes and beliefs about aging. For the NEG subscale, we predicted that higher scores would be associated with more negative self-perceptions of aging, including more *negative beliefs* about the impact of aging (APQ4) and about how much control one has over certain aspects of aging (APQ7), more *negative emotional responses* to aging (APQ5), and a stronger conscious awareness of one’s age (APQ1 and APQ2). On the other hand, we predicted higher NEG scores to reduce agreement with APQ dimensions that are related to more positive views on aging, including positive beliefs about the impact of aging (APQ3) and how much control one has over certain aspects of aging (APQ6). The negative APQ dimensions were expected to reduce quality of life and mental well-being, and enhance depressive and anxiety symptoms, feelings of loneliness and perceived stress; the positive APQ dimensions were expected to show opposite effects. 

Altogether, for the NEG subscale these predictions resulted in two types of mediation effects for self-perceptions of aging: (1) a negative mediation effect for quality of life and mental well-being, and (2) a positive mediation effect for depression, anxiety, loneliness, and perceived stress.

Reversed relationships were expected between POS scores and the APQ dimensions, although less strongly than observed for the negative ageism subscale. Consequently, for the POS subscale, we predicted (1) a positive mediation effect for quality of life and mental well-being, and (2) a negative mediation effect for depression, anxiety, loneliness, and perceived stress.

## 6. Materials and Methods (Analysis 2)

### 6.1. Participants

Participants for this analysis were drawn from the same pool of individuals as in Analysis 1. Specifically, the second group of 500 participants that completed the full battery of questionnaires and tests was included here (see Table 1 for demographic characteristics). 

### 6.2. Materials

#### 6.2.1. Perceived Aging Questionnaire (PAQ-8)

See Section 2 for details.

#### 6.2.2. Additional Measures

*Quality of Life.* The World Health Organization Quality of Life (WHOQOL)-OLD instrument [71,72] (*α* = 0.87) was used to assess QoL based on six subscales of four items each: (1) sensory abilities, (2) autonomy, (3) satisfaction with past, present, and future activities and achievements in life, (4) social participation, concerns, (5) worries and fears about death and dying, and (6) being able to have personal and intimate relationships. All 24 items were scored on a 5-point Likert Scale, with different wordings, and summed to a total QoL score. Some items were reverse scored prior to summation, such that higher scores were indicative of better quality of life.

*Mental well-being.* The 14-item Warwick Edinburgh Mental Wellbeing Scale (WEMWBS; [73,74]; *α* = 0.87) was used to measure subjective well-being and psychological functioning. All 14 items, all addressing positive aspects of mental health, were scored on a 5-point Likert scale (1 = never, 2= barely, 3 = sometimes, 4 = often, 5 = always) and summed to a total of 14 to 70. Higher scores indicated better mental well-being.

*Depression.* The Centre of Epidemiological Studies Short Depression Scale (CES-D-10; [75]; *α* = 0.82) was used as a measure of depressive symptomatology. Responses were rated on a 4-point scale, ranging from 0 (less than one day) to 3 (5–7 days) and summed to a total score of 0 to 30. Prior to summation, two positively formulated items were reverse scores. Higher scores were indicative of more depressive symptoms during the past week. 

*Anxiety.* Anxiety was measured using the 7-item anxiety subscale of the Hospital Anxiety and Depression Scale (HADS-A; [76,77]; *α* = 0.84). Participants indicated to what extent they experienced feelings of restlessness, tenseness or panic over the past two weeks on a Likert scale ranging from 0 (rarely or never) to 3 (mostly or always). Scores were summed to a total ANX score (0–21), with higher scores indicating more anxiety symptoms. 

*Loneliness.* The 11-item Loneliness Scale (LS), as developed by de Jong-Gierveld and van Tilburg [78] was used to assess overall levels of loneliness (*α* = 0.86). The scale consists of 11 items, with six items being formulated negatively (emotional subscale) and 5 items positively (social subscale). Possible answers are: “yes!”, “yes”, “more or less”, “no”, “no!”. Scores were calculated by counting the number of neutral and positive answers on the positively formulated items, and the number of neutral and negative answers on the negatively worded items. Higher sum scores (0–11) reflected more loneliness. 

*Perceived stress.* The perception of stress during the last two weeks was assessed with the Perceived Stress Scale [79,80] (*α* = 0.86). The scale consists of 6 negatively and 4 positively worded items, which were all scored on a 5-point Likert scale ranging from 1 (never) to 5 (very often). Scores were summed (positively worded items were reverse coded) across all scale items to a total of 0 to 40 points. Higher PS scores indicated higher levels of perceived stress.

*Self-perceptions of aging.* The shortened version of the Aging Perceptions Questionnaire (APQ-S) was used to assess older adults’ perceptions of aging [62]. This scale consists of 21 items and comprises seven subscales. The chronic timeline (APQ1; *α* = 0.77) subscale encompasses perceptions of aging that are chronic and constantly present (e.g., ‘I am aware of my age’), and the cyclical timeline (APQ2; *α* = 0.75) subscale refers to perceptions that fluctuate (e.g., ‘I go through phases of feeling old’). The positive consequences (APQ3; *α* = 0.75) and negative consequences (APQ4; *α* = 0.84) subscales concern beliefs about the impact of aging on various life domains, either positive (e.g., ‘As I get older, I get wiser’) or negative (e.g., ‘Getting older restricts the things that I can do’). The *emotional representations* (APQ5; *α* = 0.72) subscale refers to negative emotional responses to aging (e.g., ‘I get depressed when I think about getting older’). Finally, two subscales relate to beliefs about the extent to which one has control over various aspects of aging, both positive (APQ6, e.g., ‘The quality of my social life in later years depends on me’; *α* = 0.74) or negative (APQ7, e.g., ‘Slowing down with age is not something I can control’; *α* = 0.64) Answers are given on a 5-point scale, ranging from strongly disagree (1) to strongly agree (5), and items are summed to subscale scores, ranging from 3–15. 

### 6.3. Procedure

#### 6.3.1. Confirming Dimensionality

A Confirmatory Factor Analysis (CFA) with maximum-likelihood and *promax* rotation was performed to determine whether the established factor structure in Analysis 1 could be replicated in a second sample [81]. As in Analysis 1, criteria for goodness-of-fit indices were set to at least 0.9 for TLI and confirmatory fit index (CFI), and less than 0.05 for RMSEA and SRMR. Again, a RMSEA between 0.05 and 0.08 was considered acceptable. In addition, inter-item correlations and item-total (subscale) correlations were also calculated and compared to the metrics of Analysis 1. 

#### 6.3.2. Reliability and Scale Statistics

Internal consistency of the (subscales of the) PAQ was assessed using Cronbach’s alpha, and sum scores were calculated. Outcomes were compared to those observed in Analysis 1. 

#### 6.3.3. Validity

Validity was examined by establishing the relationship between the NEG and POS ageism subscores of the PAQ-8, with age, quality of life, mental well-being, depression, anxiety, loneliness, and perceived stress, and various dimensions of the APQ. Because of the large sample size, the sampling distribution of the variables was assumed to be normal. Histograms with fitted normal curves and QQ-plots confirmed this normal distribution. 

First, Pearson’s correlation coefficients for the relationships between PAQ subscale scores and the selected variables were calculated. Subsequently, to provide additional validity, a total of 12 mediation analyses were performed (in SPPS 26, PROCESS toolbox; [82]) to determine the total effects (c), direct effects (c’) and indirect effects (a*b), through one’s self-perceptions of aging between both POS and NEG scores, and quality of life, mental well-being, depression, anxiety, loneliness, and perceived stress. In line with our hypotheses, we expected to find a significant indirect effect for all models, with at least a partial mediation. 

## 7. Results (Analysis 2)

### 7.1. Confirming Dimensionality

The CFA revealed that the two-factor model had a good fit, with CFI = 0.976, TLI = 0.965, RMSEA = 0.049 [95% CI 0.029–0.070], and SRMR 0.034 (see Appendix A for factor loadings). The two factors were not significantly correlated, *r* = 0.08, *p* = 0.152. The item-total (subscale) correlations for the PAQ-8 were comparable with those in Analysis 1 (see Table 5), and the inter-item correlations for the subscales were between the expected range of 0.15 and 0.50, with only one exception (see Appendix A). 

### 7.2. Reliability and Scale Statistics

Cronbach’s alpha for the PAQ-8 and positive and negative subscales were 0.67 [0.62–0.71], 0.71 [0.67–0.75], and 0.79 [0.76–0.82], respectively. Comparisons of PAQ, POS, and NEG scores between the samples of Analysis 1 and 2 revealed no significant differences for POS (sample 2: *M* = 9.58, SD = 2.30) scores, *t*(998) = −0.08, *p* = 0.934. However, overall PAQ scores (sample 2: *M* = 16.8, SD = 3.51) were higher in the sample of Analysis 1, *t*(998) = 2.13, *p* = 0.034, due to higher NEG scores (sample 2: *M* = 7.20, SD = 2.52) in this sample, *t*(998) = 2.88, *p* = 0.004. 

### 7.3. Validity

Pearson’s correlation coefficients are shown in Table 6. Mediation diagrams, with all statistics, are shown in Appendix A. Below, the unmediated and direct effects, as well as indirect effects are reported. 

#### 7.3.1. Unmediated Effects and Effects on the Mediator

The unmediated effects (*c-paths)* of POS and NEG scores on quality of life, mental well-being, depression, anxiety, loneliness, perceived stress, as well as on the different dimensions of one’s self-perception of aging (APQ; *a-paths*), are reported in Table 7. Unmediated effects were significant for all pairs of variables, except for POS scores and perceived stress. Different associations with APQ dimensions were found for POS and NEG scores, generally in line with the idea that positive ageism leads to more positive perceptions of aging and negative ageism evokes negative thoughts and attitudes towards one’s own aging.

#### 7.3.2. Direct and Indirect Effects

Self-perceptions of aging were found to reliably mediate the relationship between NEG scores as independent variables and quality of life, mental well-being, depression, anxiety, loneliness and perceived stress as outcome variables (see Table 8; *a*b*). Higher levels of perceived negative ageism led to more negative perceptions of aging and thereby caused more unfavorable (mental) health outcomes. Interestingly, self-perceptions of aging were found to completely mediate the relationship between NEG and mental well-being (no direct (c’-path) effect remaining), indicating that the entire effect of NEG scores on mental-wellbeing is transmitted through self-perceptions of aging.

Similarly, one’s own perceptions of aging were found to mediate the relationship between POS scores and the outcome variables. Complete mediation was found for quality of life and perceived stress. For the latter, an unmediated effect was not observed, yet the significant indirect effect suggests that an effect of POS on perceived stress, via self-perceptions of aging, might be present after all.

Similar effects were observed when all paths (direct and indirect, for both NEG and POS, and for all outcome variables of interest) were evaluated using a path-analysis approach in R (using the *Lavaan* Package, with Maximum Likelihood estimation; see Appendix A; the R code is included in Appendix A). This shows that our results are robust.

## 8. Interim Discussion (Analysis 2)

In Analysis 2, the factor structure of the PAQ-8 that was established in Analysis 1 was confirmed in an independent sample (*n* = 500). Moreover, the subscales of the PAQ were found to hold good validity, as the relationships found for the NEG and POS subscales were in line with theoretical hypotheses and existing knowledge of the concept of ageism. The levels of perceived negative and positive ageism were both positively associated with age, albeit less strongly for the positive subscale, which is in line with the idea that ageism increases with age [32]. Our scale is also consistent with the hypothesis of a negative effect of *negative* ageism on mental health (e.g., [15,18]) by showing a negative relationship between the level of perceived negative ageism and both quality of life and mental well-being, and positive relationships with depression, anxiety, loneliness, and perceived stress (e.g., [42,43]) (i.e., unmediated effects). Moreover, opposite patterns for the level of perceived positive ageism confirm that positive ageism may indeed have opposite consequences for mental well-being, and that the PAQ-8 can reliably assess negative and positive forms of ageism. 

The most convincing evidence for the *subscale validity* comes from the relationships that were observed with the seven APQ subscales. That is, we found specific patterns of relationships for both positive and negative subscales that are in line with the idea that negative ageism generally has negative consequences for one’s self-perceptions of aging (e.g., [17,18]), whereas positive ageism may actually improve one’s attitude towards one’s own aging (e.g., [22,41]). Specifically, those who reported high levels of *perceived negative ageism* also scored higher on the APQ subscales that relate to *negative beliefs* about the impact of aging and how much control a person has over certain aspects of aging. Relationships with *perceived positive ageism* for these APQ subscales were nonsignificant. Similarly, the APQ subscales that relate to *positive beliefs* about the impact of aging and how much control a person has over certain aspects of aging correlated positively with the level of *perceived positive ageism*. These APQ subscales were not associated with the level of *perceived negative ageism*. Finally, the APQ subscale that relates to *negative emotional responses* to aging was *positively* related with the level of *perceived negative ageism,* and *negatively* correlated (albeit only weakly) with the level of *perceived positive ageism*. Both chronic and cyclical awareness of the aging timeline were positively related to levels of *perceived negative ageism*, meaning that those who perceive more negative ageism are also more aware of their age and aging timeline, either constantly or in phases. The level of perceived positive ageism was negatively related to individuals’ reported cyclical awareness, but not chronic awareness. 

Additionally, in line with the theoretical framework that perceived ageism can impact mental health due to the assimilation of beliefs and perceptions about aging, the relationships between NEG and POS scores and the outcome variables were all mediated by self-perceptions of aging. This may constitute the most compelling evidence for good validity of both the negative and positive subscale of the PAQ-8. For some mediation models—NEG on mental well-being, POS on quality of life and perceived stress—the indirect (mediated) effect fully explained the relationship between the independent and dependent variable. This suggests that the deleterious effects of negative ageism on mental well-being, and beneficial effects of positive ageism on quality of life and perceived stress, are due to the assimilation of the beliefs and attitudes related to the stereotypes, prejudices, and discrimination, thereby shaping one’s perceptions of aging. 

In most cases, the relationships between the PAQ-8 subscales and the (mental) health outcome variables were only partially mediated by self-perceptions of aging. This is consistent with the idea that there are multiple pathways via which perceived ageism can affect functioning and (mental) health [11]. For instance, an additional pathway to loneliness may be that frequent social rejection inclines older adults to avoid and/or withdraw from social participation, thereby promoting feelings of loneliness [83]. Another potential mechanism linking negative ageism to depressive symptoms is the negative emotional reactions (i.e., anger, sadness, powerlessness) that are evoked by disturbing experiences [84].

Indeed, it may also be that some mediating factors are not independent, but act as transmitters prior to or after other mediators in a longer chain. For instance, Han and Richardson [17] showed that perceived ageism can improve negative self-perceptions of aging, thereby decreasing the sense of purpose of life, which subsequently leads to more depressive symptoms. To unravel unknown pathways from ageism to mental health, exploratory network analysis may prove highly informative, as it allows for the inclusion of many variables and the visualization of their interplay (see e.g., [85]). 

In summary, Analysis 2 provided convincing evidence that the PAQ-8, and its subscales in particular, have good validity and can be used to reliably assess perceived ageism. 

## 9. General Discussion

To increase the potential for successful aging, age barriers and ageism need to be combatted. However, in order to do so, it is important to further improve our understanding of the effects of ageism and be able to evaluate the effectivity of various attempts to reduce ageism. This calls for a reliable and comprehensive measure to assess the *subjective experience* of all forms and dimensions of ageism. In this study, we used a rational-empirical approach in the construction and validation of the Perceived Ageism Questionnaire (PAQ) for older adults. This scale was developed with the intention to reflect all three dimensions of ageism (stereotypes, prejudices, and discrimination), and to include both positive and negative forms. In Analysis 1, the 8-item PAQ (PAQ-8) showed good psychometric and structural properties, and revealed a reliable two-factor structure, distinguishing positive (POS) and negative (NEG) forms of ageism, which was confirmed in Analysis 2. Additionally, in Analysis 2, both subscales demonstrated high validity, as evidenced by meaningful relationships with indices of quality of life, mental well-being, mental health and self-perceptions of aging. Below, we critically evaluate the overall findings, discuss the implications, and provide suggestions for future studies in which the PAQ-8 could help to obtain relevant, and sometimes urgently needed, insights. 

### 9.1. Contribution of the Two-Dimensional PAQ

As has often been pointed out, a major hindrance for ageism research is that studies use different definitions and measures of ageism (e.g., [11,86]). Hence, as a matter of priority, the WHO and its collaborators are currently developing a comprehensive scale to assess the prevalence of ageism in society [6]. Our scale development effort, resulting in the PAQ-8, provides a useful complement to this scale and can help to gather more reliable, standardized and comparable data of the level, role and impact of *ageism as perceived and experienced* by older individuals themselves. The PAQ-8 can be used to examine the impact of ageism in society on the individual level, and disentangle potential protective and risk factors that can affect one’s level of resilience to the consequences of ageism (also see Section 9.2).

Unlike other existing scales that have previously been used to assess *perceived ageism*, the PAQ-8 covers all dimensions of ageism (stereotypes, prejudices and discrimination) and can be used to assess the differential impacts of positive and negative forms of ageism, as well as the efficacy of interventions targeting one or both of these forms of ageism. Research on the effects of positive ageism is scarce. To validate the perceived positive ageism subscale of the PAQ, we relied on previous studies revealing positive effects of positive ageism (e.g., [42,43]). In line with our hypotheses, we observed positive associations with quality of life, mental well-being, and negative associations with depression, anxiety, loneliness, and perceived stress.

Our findings are also in line with earlier evidence showing that negative forms of ageism tend to exert relatively stronger effects on negative outcomes than positive forms of ageism have on positive outcomes [27]. As some studies have reported contradictory effects, with positive age stereotypes making older adults feel older [3], it is important for future research to extend this line of research and evaluate the circumstances under which exposing individuals to positive forms of ageism may be beneficial and/or even diminish or reverse the negative consequences of perceived negative age stereotypes, prejudices, and discrimination (e.g., [22]). The PAQ-8 lends itself well to answering such questions. Moreover, the PAQ-8 can be used to unravel specific pathways through which perceived negative or positive ageism affect physical and mental health in the long run. This would greatly benefit the field, as the majority of existing studies rely on cross-sectional data, which does not allow for testing causality.

### 9.2. Level of Perceived Ageism and Individual Differences

On average, levels of perceived ageism, and in particular negative forms, appeared to be relatively low. This does not concur well with the general trends and discourse on *manifested* ageism, showing that the prevalence of ageism in society has largely increased over the past century [87] and that ageism is widespread and one of the most common forms of prejudices, stereotypes, and discrimination in our society [6,10,88]. However, ageism also often goes largely unrecognized, partially due to the strong internalization of beliefs and attitudes regarding older adults and aging from experiences early in life onwards. Consequently, even if individuals approach older adults on the basis of stereotypes or prejudices, older individuals may not always consciously perceive this as discriminatory because such ageist attitudes are fully integrated into their existence and expectations of the world. This would result in relatively lower (subjective) scores on the PAQ, while levels of ageism in society are nonetheless high.

Despite the relatively low levels of perceived ageism, considerable individual differences among individuals were present. Indeed, individual differences in both perceived negative and positive ageism were partially explained by age [32], with higher levels of perceived ageism being observed among those high in age. However, even younger individuals (≤67 years old) sometimes reported relatively high levels of perceived negative and positive ageism. This does not only confirm that discrimination, stereotyping and prejudices against older adults on the grounds of their age can occur at a relatively young age (e.g., on the labour market; [89]), but also suggests that other characteristics such as gender, education, marital status, wealth, employment status [90], as well as culture, religion, or neighborhood (e.g., percentage of older residents) may also affect the level of ageism as experienced by older adults [91,92]. For instance, low- and middle-income countries are associated with a higher prevalence of ageism, and individualistic cultural values foster more negative attitudes towards older adults [93], which is most likely reflected in older adults’ perceptions of ageism as well. On the other hand, strong religious commitments can reduce death anxiety [94], and thereby potentially diminish perceived ageism (i.e., death anxiety is known to give rise to more negative views of aging and ageist attitudes; see [95,96,97]). Future research could use the PAQ-8 to determine which factors indeed drive individual differences in experiences and perceptions of ageism. 

Another source for individual differences in perceived ageism may be that some individuals are biased and inclined to perceive the world more negatively or positively. Indeed, our mediation models in Analysis 2 showed that perceptions of aging mediate the relationship between NEG/POS and mental health outcomes, in line with the idea that frequent exposure to ageism can lead to the internalization of the negative or positive beliefs and attitudes, affecting one’s perceptions of aging and thereby impacting mental functioning. However, an alternative, albeit speculative conjecture is that increased mental health problems evoke more negative thoughts about aging, thereby causing individuals to view the world and everyday events more negatively. Similarly, decreased mental health problems could give rise to more positive attitudes towards aging and thereby more positive perceptions of the environment. This may subsequently promote a reinforcing loop that maintains either favorable, in case of POS, or unfavorable, in case of NEG, health outcomes. In line with the idea that negative emotions can elicit more negative thoughts about aging, it has been suggested that perceived isolation can make older adults more dissatisfied with the quality of their relationships, which increases their pessimism and thereby promotes negative attitudes towards aging [98]. Furthermore, an earlier prospective study has shown that depressive symptoms can increase the likelihood of perceived age discrimination [99]. This finding is explained using the *cognitive triad*, which suggests that depression can incur a negative view of oneself, the world, and one’s future [100]. According to Ayalon [99] individuals may therefore be more likely to perceive neutral events as discriminatory, and experience more negative emotional reactions from them. Alternatively, depressed individuals may mistakenly interpret their somatic symptoms as signs of age-related decline [101], thereby affecting their self-perceptions of aging and responding more negatively to (presumably) ageism. It is important to extend this line of research to the field of overall ageism (including stereotypes, prejudices, and discrimination) and examine the role of other mental health problems. As acknowledged by Ayalon [99] there might be less societal incentive to show that mental health problems can precede perceived ageism, as it does not help to advocate for more equal rights, but it is important to get a better idea of the (potential) subjective nature of perceived ageism. Again, the PAQ-8 provides a comprehensive tool to examine such interplays. 

In addition to explanatory factors for individual differences in *perceived* ageism, it is also of high importance to disentangle which factors may exacerbate or alleviate (i.e., moderate) *the impact of* perceived negative ageism on older adults’ quality of life, mental well-being, and mental health outcomes, either directly or indirectly (via mediating factors). Previous research has shown that some factors may help buffer the detrimental effects of negative ageism, or self-perceptions of aging, including optimism, body esteem, coping and high levels of flexible goal adjustment [84,102,103,104] (see for review: [86]), whereas other factors (e.g., anxiety about aging [105], or young age [15]) may promote more negative consequences. Using the NEG subscale of the PAQ-8, we can further extend our knowledge of the factors that can mitigate the physical and mental health effects of ageism, which is a prerequisite for developing effective strategies that aim to diminish maladaptive consequences of (perceived) ageism. 

### 9.3. Potential Limitations 

One important limitation to consider with regards to our scale development effort is the limited size of our initial pool of items. In total, nine items were considered to be relevant to the ageism construct and were therefore subjected to psychometric testing, resulting in an eight-item PAQ with good psychometric properties and high validity. This suggests that the included items provided sufficient depth, specificity, or bandwidth. However, commonly, a larger pool of items is considered recommended [106], leaving open the possibility that we have missed some critical examples of ageism that can meaningfully explain individual differences in the level of perceived ageism. 

As the prevalence of factors that influence older adults’ perception of ageism, as well as the subsequent impact on physical and mental health, can sometimes be ingrained in specific countries or cultures (e.g., some countries/cultures are more optimistic or religiously committed than others; some countries are individualistic, whereas others are collectivistic), simply generalizing our findings to other samples might not be entirely warranted. For some countries or cultures, the examples included in the PAQ-8 may not represent the most commonly encountered forms of stereotypes, prejudices, or discrimination. Hence, future research should test how the reliability and validity of the PAQ-8 holds in different groups and cultures and take into account potential subgroup-related factors that may drive differences in responding. 

Finally, as the PAQ-8 quantifies the level of perceived ageism throughout the previous year, this may introduce some limitations with regards to the applicability of the scale in longitudinal designs. Potential changes in perceived ageism (e.g., due to an intervention or campaign to reduce (perceived) ageism) can only be assessed approximately one year later (some overlap may not be a problem). Future research should evaluate whether the PAQ-8 could also be used to reliably assess the level of perceived ageism (and changes therein) throughout shorter periods of time (e.g., two or four weeks).

## 10. Conclusions

In summary, the PAQ-8 provides a reliable and comprehensive tool to quantify the levels of ageism as experienced by older adults in the Dutch population. The scale can be used to further examine relevant pathways from ageism to physical and mental health outcomes, as well as potential attempts to combat ageism in our society. Since prejudices and stereotypes are (often) strongly ingrained [6], aiming for short-term changes in ageist beliefs and attitudes within individuals may be unrealistic. However, interventions that focus on improving awareness of the prevalence and consequences of ageism (particularly among younger individuals) could help to encourage individuals to alleviate the manifestation of inter-personal ageism, and thereby reduce the level of ageism as experienced and perceived by older individuals. The PAQ-8 is well suited to capture such subjective changes. Eventually, this may also help to rewire our thinking patterns and overcome the implicit age-related biases that are formed through the chronic lifetime exposure to negative views about older adults.

## Figures and Tables

**Figure 1 ijerph-19-08792-f001:**
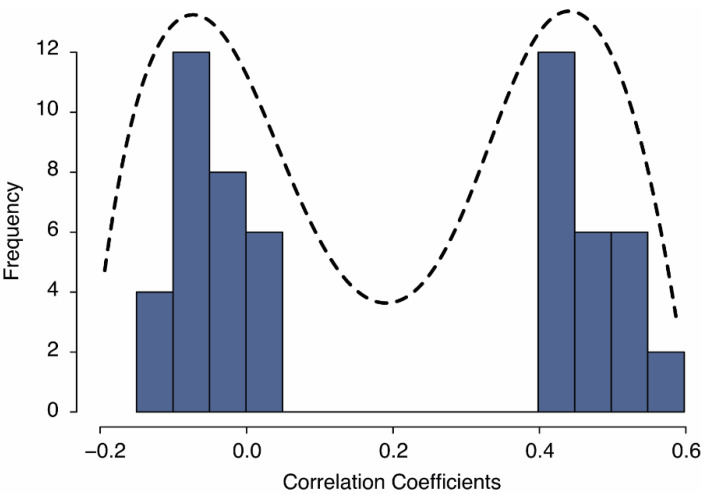
The bi-modal inter-item correlation frequency distribution for the two-dimensional PAQ-8.

**Table 1 ijerph-19-08792-t001:** Demographic characteristics of samples included in Analysis 1 and 2.

Analysis	1	2
	N = 500	N = 500
^a^ Age: M (SD), range	69.0 (7.16), 55–93	67.7 (7.45), 55–92
^b^ Gender: % female	65%	70%
Male	175	149
Female	324	351
Other	1	0
^c^ Education: % high ^d^	82%	87%
Low	3	1
Middle	88	65
High	409	434

^a^ Note that although the mean age of the sample of Analysis 1 is slightly higher than the sample of Analysis 2, *t*(998) = 2.82, *p* = 0.005, we consider the difference negligible. ^b^
*Χ*(2) = 4.17, *p* = 0.125. ^c^
*Χ*(2) = 5.20, *p* = 0.074. ^d^ According to the Dutch Verhage [46] scale, categories 6 and 7 reflect a high education level.

**Table 2 ijerph-19-08792-t002:** Descriptive statistics (means, SD) and corrected item-total scale correlations of Analysis 1 for the PAQ-9 and PAQ-8.

Item		M	SD	Corrected Item-Total Scale Correlations					
				PAQ-9			PAQ-8		
How often in the past year have you had the feeling that...				PAQ	NEG	POS	PAQ	NEG	POS
1	... people approach you as if you are a child because of your age?	1.29	0.61	0.39	0.57		0.37	0.58	
2	... you are not being listened to and/or your opinion advice is not taken seriously because of your age?	1.52	0.73	0.38	0.56		0.36	0.57	
3	... people value your advice and contribution to a conversation because of your life experience?	3.40	0.82	0.21		0.62	0.22		0.62
4	... people hold negative prejudices or exaggerated stereotypes (e.g., weak, vulnerable, dull, slow) about you because of your age?	1.66	0.81	0.42	0.59		0.40	0.60	
5	... people assume that you are wise and sensible because of your age?	2.93	0.91	0.26		0.61	0.28		0.61
6	... people unjustly treat you as if you are less capable (mentally or physically) because of your age?	1.76	0.80	0.48	0.63		0.44	0.63	
7 ^a^	... people incorrectly assume that you cannot hear well because of your age?	1.45	0.73	0.40	0.43				
8	... people think disparagingly (with little or no dignity) about your contribution to society because of your age?	1.46	0.71	0.44	0.63		0.38	0.59	
9	... people see you as a meaningful part of society precisely because of your age (e.g., babysitting (grand)children or volunteer activities)?	3.24	1.09	0.26		0.53	0.26		0.53

*Note.* NEG: perceived negative ageism subscale, POS: perceived positive ageism subscale. In this study, we used a Dutch version (see Appendix A). ^a^ This item was removed on the basis of the results of the Exploratory Factor Analysis and is not included in the final PAQ-8.

**Table 3 ijerph-19-08792-t003:** PAQ item factors loadings based on Exploratory Factor Analysis (Analysis 1).

Item	Factor Loadings			
	PAQ-9		PAQ-8	
	NEG	POS	NEG	POS
1	**0.63**	−0.06	**0.65**	−0.03
2	**0.63**	−0.06	**0.64**	−0.04
3	−0.07	**0.78**	−0.07	**0.77**
4	**0.68**	−0.01	**0.69**	0.02
5	0.01	**0.76**	0.02	**0.77**
6	**0.73**	0.03	**0.73**	0.06
7	**0.49**	0.10		
8	**0.70**	−0.03	**0.67**	−0.01
9	0.04	**0.62**	0.04	**0.62**

*Note*. NEG: perceived negative ageism subscale, POS: perceived positive ageism subscale. For each item, the highest factor loading is in bold.

**Table 4 ijerph-19-08792-t004:** Goodness-of-fit indices for the Exploratory Factor Analyses (Analysis 1).

	TLI	RSMEA	RMSR	BIC
PAQ-9	0.95	0.061	0.033	−64.20
PAQ-8	0.97	0.051	0.025	−51.07

*Note*. Tucker Lewis Index (TLI); Root Mean Square Error of Approximation (RMSEA); Standardized Root Mean Squared Residual (SRMR); Bayesian Information Criterium (BIC; note that unlike the other indices, lower BIC values indicate better goodness of fit).

**Table 5 ijerph-19-08792-t005:** Descriptive statistics (means, SD) and corrected item-total scale correlations of Analysis 2 for the PAQ-8.

Item	M	SD	PAQ-8	NEG	POS
1	1.21	0.50	0.37	0.53	
2	1.50	0.75	0.48	0.60	
3	3.38	0.89	0.32		0.55
4	1.53	0.76	0.42	0.61	
5	2.98	0.88	0.37		0.58
6	1.60	0.73	0.35	0.56	
7	-	-	-	-	-
8	1.36	0.64	0.40	0.58	
9	3.22	1.10	0.29		0.48

*Note*. NEG: perceived negative ageism subscale, POS: perceived positive ageism subscale.

**Table 6 ijerph-19-08792-t006:** Pearson’s correlation coefficients for the relationship between perceived negative (NEG) and positive (POS) ageism subscales and variables of interest.

Construct	NEG (r)	POS (r)
Age ^a^	0.14 **	0.11 *
Depression	0.30 ***	−0.21 ***
Anxiety	0.29 ***	−0.17 ***
Loneliness	0.21 ***	−0.20 ***
Perceived stress	0.30 ***	−0.08
Quality of Life	−0.36 ***	0.16 ***
Mental well-being	−0.20 ***	0.28 ***
APQ1 (timeline chronic)	0.15 ***	0.04
APQ2 (timeline cyclical)	0.31 ***	−0.13 **
APQ3 (consequences positive)	−0.08	0.21 ***
APQ4 (consequences negative)	0.25 ***	−0.08
APQ5 (emotional representations)	0.32 ***	−0.11 *
APQ6 (control positive)	−0.08	0.13 **
APQ7 (control negative)	0.15 ***	−0.07

*Note*. * *p* < 0.05, ** *p* < 0.01, *** *p* < 0.001. ^a^ When dividing participants into two age groups (≤67 and >67 years old), we observed that the range of NEG and POS scores was highly similar among groups (5–16 versus 5–19 for NEG, and both 3–15 for POS). See Appendix A, for more details.

**Table 7 ijerph-19-08792-t007:** Unmediated effects of perceived negative (NEG) and positive (POS) ageism scores on (mental health) outcome variables, and their relationships with different dimensions of self-perceptions of aging.

Construct	NEG (r)			POS (r)		
	B	SE	95% CI	B	SE	95% CI
Depression	0.538 ***	0.077	[0.39, 0.69]	−0.405 ***	0.086	[−0.57, −0.23]
Anxiety	0.379 ***	0.055	[0.27, 0.49]	−0.236 ***	0.062	[−0.36, −0.11]
Loneliness	0.252 ***	0.052	[0.15, 0.35]	−0.261 ***	0.057	[−0.37, −0.15]
Perceived stress	0.617 ***	0.089	[0.44, 0.79]	−0.180	0.102	[−0.38, 0.02]
Quality of Life	−1.350 ***	0.155	[−1.66, −1.04]	0.670 ***	0.180	[0.32, 1.02]
Mental well-being	−0.470 ***	0.103	[−0.67, −0.27]	0.715 ***	0.111	[0.50, 0.93]
APQ1 (timeline chronic)	0.141 ***	0.042	[0.06, 0.22]	0.039	0.047	[−0.05, 0.13]
APQ2 (timeline cyclical)	0.295 ***	0.041	[0.22, 0.37]	−0.134 **	0.046	[−0.22, −0.05]
APQ3 (consequences positive)	−0.055	0.032	[−0.12, 0.01]	0.164 ***	0.035	[0.10, 0.23]
APQ4 (consequences negative)	0.235 ***	0.042	[0.15, 0.32]	−0.082	0.047	[−0.17, 0.01]
APQ5 (emotional representations)	0.250 ***	0.033	[0.19, 0.31]	−0.097 *	0.038	[−0.17, −0.02]
APQ6 (control positive)	−0.056	0.030	[−0.12, 0.00]	0.099 **	0.033	[0.03, 0.16]
APQ7 (control negative)	0.114 ***	0.034	[0.05, 0.18]	−0.056	0.037	[−0.13, 0.02]

*Note*. APQ: Aging Perceptions Questionnaire, comprising 7 subscales. B represents unstandardized regression weights. Square brackets are used to enclose the lower and upper limits of a confidence interval (CI). * *p* < 0.05, ** *p* < 0.01, *** *p* < 0.001.

**Table 8 ijerph-19-08792-t008:** Direct and indirect effects, via self-perceptions of aging, of the mediation models.

		NEG (r)			POS (r)		
		B	SE	95% CI	B	SE	95% CI
Depression	Direct	0.276 ***	0.075	[0.13, 0.42]	−0.276 ***	0.080	[−0.43, −0.12]
	Indirect	0.262	0.049	[0.18, 0.36]	−0.129	0.046	[−0.22, −0.04]
Anxiety	Direct	0.216 ***	0.055	[0.11, 0.32]	−0.162 **	0.059	[−0.28, −0.05]
	Indirect	0.163	0.033	[0.10, 0.23]	−0.074	0.031	[−0.13, −0.01]
Loneliness	Direct	0.130 *	0.054	[0.02, 0.24]	−0.191 ***	0.056	[−0.30, −0.08]
	Indirect	0.121	0.026	[0.07, 0.18]	−0.071	0.025	[−0.12, −0.02]
Perceived stress	Direct	0.358 ***	0.090	[0.18, 0.53]	−0.023	0.096	[−0.21, 0.17]
	Indirect	0.260	0.050	[0.17, 0.36]	−0.157	0.052	[−0.26, −0.06]
Quality of Life	Direct	−0.646 ***	0.135	[−0.91, −0.38]	0.218	0.145	[−0.07, 0.50]
	Indirect	−0.704	0.110	[−0.93, −0.49]	0.452	0.131	[0.19, 0.70]
Mental well-being	Direct	−0.149	0.098	[−0.34, 0.04]	0.455 ***	0.102	[0.26, 0.66]
	Indirect	−0.320	0.062	[−0.45, −0.20]	0.260	0.064	[0.14, 0.39]

*Note*. NEG: perceived negative ageism subscale, POS: perceived positive ageism subscale. B represents unstandardized regression weights. Square brackets are used to enclose the lower and upper limits of a confidence interval (CI). ** p* < 0.05, *** p* < 0.01, **** p* < 0.001.

## Data Availability

The datasets presented in this article are not readily available because the datasets used and/or analyzed for the current study will only be made publicly available after completion of the overarching project, and will until that time only be available from the corresponding author on collaboration basis upon reasonable request. Requests to access the datasets should be directed to L.P.B., l.p.brinkhof@uva.nl.

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
