# Peer review of "The Subjective Experience of Ageism: The Perceived Ageism Questionnaire (PAQ)"

_ijerph, 2022, doi:10.3390/ijerph19148792_

Round 1

Reviewer 1 Report

The authors developed PAQ-8, which could help to gather more standardized data of the level, role, and impact of perceived ageism. 

 In this study, the authors provide the PAQ-8 the quantify the levels of ageism as experienced by older adults in the Dutch population. The scale can be used to further examine relevant pathways from ageism to physical and mental health outcomes, as well as potential attempts to combat ageism in our society. Major concerns:

  1. the study is descriptive in general.
  2. Many papers have been published regarding the scale of methodological critique for ageism. what is the innovation of this study compared to others?
  3. Please describe the pros and cons of the PAQ-8 in detail.

Author Response

We thank the reviewer for her/his constructive questions, which help bring our paper further into focus.

As of yet, no validated scale is available that can reliably assess the subjective experience of all forms and dimensions of ageism. The PAQ-8 offers a novel, comprehensive tool to quantify the level of perceived and experienced ageism.

As to the pros: As described in the introduction and discussion sections, the PAQ-8 is the first scale that combines all dimensions of ageism, and includes both negative and positive forms of ageism. We have included some additional sentences to our discussion section (603 – 609) to emphasize this even further, and also mention the clinical/social relevance of the PAQ-8:

“Our scale development effort, resulting in the PAQ-8, provides a useful complement to this scale and can help to gather more reliable, standardized and comparable data of the level, role and impact of ageism as perceived and experienced by older individuals themselves. The PAQ-8 can be used to examine the impact of ageism in society on the individual level, and disentangle potential protective and risk factors that can affect one’s level of resilience to the consequences of ageism (also see section 8.2).

Unlike other existing scales that have previously been used to assess perceived ageism, the PAQ-8 covers all dimensions of ageism (stereotypes, prejudices and discrimination) and can be used to assess the differential impacts of positive and negative forms of ageism, as well as the efficacy of interventions targeting one or both of these forms of ageism.”

The cons of the PAQ-8 have been described in section 8.3 and relate to the limited size of our initial pool of items, as well as the generalizability of the scale to other samples. In addition, we have included another potential limitation of the PAQ-8, related to the applicability of the questionnaire in longitudinal designs (721 – 727).

“Finally, as the PAQ-8 quantifies the level of perceived ageism throughout the previous year, this may introduce some limitations with regards to the applicability of the scale in longitudinal designs. Potential changes in perceived ageism (e.g., due to an intervention or campaign to reduce (perceived) ageism) can only be assessed approximately one year later (some overlap may not be a problem). Future research should evaluate whether the PAQ-8 could also be used to reliably assess the level of perceived ageism (and changes therein) throughout shorter periods of time (e.g., two or four weeks).”

Reviewer 2 Report

It is suggested to support the research in the recent publication by Kang et al. (2022) which precisely suggests creating a scale like the one included in the research whose conclusions literally state: "Study findings suggest that a new scale must be developed which applies only to assessing ageism on the basis of how older adults perceive it. In the midst of an aging population around the world, a scale that is capable of accurately estimating ageism prevalence is crucial.To adequately assess ageism, a comprehensive set of constructs, including cognitive, behavioral, and informative components, that rely on reliable and valid indicators is required For an accurate representation of the complex nature of age-based prejudices, it is important to measure both hostile and benevolent ageism as well as their explicit and implicit manifestations Finally, ageism scales must always evolve and be further refined. consider cultural differences and accurately reflect attitudes, opinions, and perceptions of older adults about aging in our current environment, amid fast-paced cultural change with new values, norms, and styles".

https://www.frontiersin.org/articles/10.3389/feduc.2022.739436/full

The minimum age of the participants is at this historical and social moment perhaps somewhat low to measure ageism (55) so it would be reasonable to discuss this aspect, provide statistical information on differences in the normative results by age groups and it would even have been advisable to in the main analyses, age would have been considered. In fact there is a significant correlation between age and ageism.

Since the responses to self-reports may be biased by response styles influenced by aspects such as mood, mental health, personality factors, social desirability, among others, it is possible that the results are contaminated by it. It is possible that a person experiencing symptoms of depression may feel more discriminated against due to a negative bias in information processing. Since there has been no control of such dimensions, the discussion should reflect this possible limitation.

The discussion would benefit from a potential paragraph about the clinical and social usefulness of the new scale compared to other existing ones.

Author Response

Comment #1

The minimum age of the participants is at this historical and social moment perhaps somewhat low to measure ageism (55) so it would be reasonable to discuss this aspect, provide statistical information on differences in the normative results by age groups and it would even have been advisable to in the main analyses, age would have been considered. In fact there is a significant correlation between age and ageism.

We thank the reviewer for raising this important point. While we understand that our age range may seem rather wide, previous studies have provided substantial evidence showing that discrimination, stereotyping and prejudices against older adults on the grounds of their age also targets relatively young seniors, for instance on the labour market (see Stypinska & Turek, 2017).

Stypinska, J., & Turek, K. (2017). Hard and soft age discrimination: the dual nature of workplace discrimination. European Journal of Ageing14(1), 49-61.

To confirm that the minimum age of 55 is reasonable, as per the reviewer’s request, we distinguished two subgroups based on age (<= 67 years old (n = 263) and > 67 years old (n = 237)) and examined their minimum and maximum values. In line with the positive correlation between ageism and age, average POS and NEG scores were higher for the oldest subgroup. However, the minimum and maximum values were relatively similar. Among individuals 67 years or younger, negative (NEG) ageism values ranged from 5 to 16 and positive (POS) ageism valued from  3 to 15. NEG and POS scores among those older than 67 years ranged from 5 to 19 and 3 to 15, respectively. This shows that, although on average NEG and POS scores are higher for those higher in age, some younger individuals do experience high levels of perceived ageism. This is also in line with previous studies stipulating that chronological age does not always proceed in sync with biological age, and that the aging process is characterized by high variability among individuals, which may also give rise to ageism at a younger age and, crucially, to elevated levels of perceived ageism.

We agree that these additional results provide some interesting insights, so we have now included these statistics in our revision (see line 472 – 474, and Supplement II) and added a remark on this in our discussion (644 – 649).

a When dividing participants into two age groups (<= 67 and > 67 years old), we observed that the range of NEG and POS scores was highly similar among groups (5 – 16 versus 5 – 19 for NEG, and both 3 – 15 for POS). See Supplement II, Table S3, for more details.”

“Despite the relatively low levels of perceived ageism, considerable individual differences among individuals were present. Indeed, individual differences in both perceived negative and positive ageism were partially explained by age [32], with higher levels of perceived ageism being observed among those high in age. However, even younger individuals (<= 67 years old) sometimes reported relatively high levels of perceived negative and positive ageism. This does not only confirm that discrimination, stereotyping and prejudices against older adults on the grounds of their age can occur at a relatively young age (e.g., on the labour market; [89]), but also suggests that other characteristics such as gender, education, marital status, wealth, employment status [90], as well as culture, religion, or neighborhood (e.g., percentage of older residents) may also affect the level of ageism as experienced by older adults [91,92].”

Additionally, we think that including age in the analyses is an interesting idea, and we agree this could provide relevant insights into the role of age in the interplay between perceived ageism, self-perceptions of aging and several (mental) health outcomes (note that previous research has already examined the moderating role of experiences of ageism and mental health; Lyons et al., 2018 – showing more negative consequences for those younger in age). However, we believe this is beyond the scope of this validation study and think that the question as to whether the (indirect) effect of perceived ageism (as measured with the PAQ) on our outcomes of interest may be moderated by age, would be more appropriate to defer to future research (perhaps in combination with other potential moderators). As we agree this is a highly interesting and important question, we have included this suggestion for future research in the discussion section in which we discuss other potential moderators (line 697).

Lyons, A.; Alba, B.; Heywood, W.; Fileborn, B.; Minichiello, V.; Barrett, C.; Hinchliff, S.; Malta, S.; Dow, B. Experiences of Ageism and the Mental Health of Older Adults. Aging Ment. Heal. 2018, 22, 1456–1464, doi:10.1080/13607863.2017.1364347.

Comment #2

Since the responses to self-reports may be biased by response styles influenced by aspects such as mood, mental health, personality factors, social desirability, among others, it is possible that the results are contaminated by it. It is possible that a person experiencing symptoms of depression may feel more discriminated against due to a negative bias in information processing. Since there has been no control of such dimensions, the discussion should reflect this possible limitation.

We agree this is an important point, and we had included a paragraph in the discussion section to emphasize this (line 659 – 688). We describe how mental health problems may evoke more negative thoughts about aging, thereby causing individuals to view the world and everyday events more negatively. In addition, we also argue that decreased mental health problems could give rise to more positive attitudes towards aging and thereby more positive perceptions of the environment. We also provide a more detailed example and discuss the findings of a previous study that showed that depressive symptomatology can increase the likelihood of perceived age discrimination (as explained by the cognitive triad). To conclude, we state that “It is important to extend this [the role of mental health problems on perceived ageism] line of research to the field of overall ageism (including stereotypes, prejudices, and discrimination) and examine the role of other mental health problems.

We believe this paragraph covers the suggestions raised here, but we will be happy to accommodate any other specific points for improvement that the reviewer brings up.

Comment #3

The discussion would benefit from a potential paragraph about the clinical and social usefulness of the new scale compared to other existing ones.

We thank the reviewer for pointing this out. We have extended section 8.1 (line 603 – 609) and subsequently refer to section 8.2, in which we also discuss some specific examples in which the PAQ may provide a reliable tool to examine relevant pathways from ageism in society, to perceived ageism and subsequently to the consequences for physical and mental health.

“Our scale development effort, resulting in the PAQ-8, provides a useful complement to this scale and can help to gather more reliable, standardized and comparable data of the level, role and impact of ageism as perceived and experienced by older individuals them-selves. The PAQ-8 can be used to examine the impact of ageism in society on the individual level, and disentangle potential protective and risk factors that can affect one’s level of resilience to the consequences of ageism (also see section 8.2).perceived ageism.

Unlike other existing scales that have previously been used to assess perceived age-ism, the PAQ-8 covers all dimensions of ageism (stereotypes, prejudices and discrimination) and Importantly, the PAQ-8 can be used to assess the differential impacts of positive and negative forms of ageism, as well as the efficacy of interventions targeting one or both of these forms of ageism.”

Reviewer 3 Report

Dear Authors,

I would expanded the introduction

Please use key word according to MeSH

Author Response

Comment #1

I would expanded the introduction

We are happy to expand our introduction in line with specific suggestions, should the reviewer present these.

Comment #2

Please use key word according to MeSH

We have changed accordingly. Next to MeSH keywords (“aging”, “mental health”, “quality of life”, etc.), we included “self-perceptions of aging” and “mental well-being” because they are critical to this paper (but not included yet in the MeSH database)

Reviewer 4 Report

I want to thank the authors and the Editorial Board for the opportunity to review the article submitted to the International Journal of Environmental Research and Public Health. The authors’ manuscript refers to a very important topic: the measurement of ageism. All statistical methods were applied correctly and the study has sufficient theoretical background. I believe that the submitted article is of high quality. Therefore, I recommend the publication of the authors’ manuscript with one major change.

Authors test indirect effects with the PROCESS model run 12 times. I highly recommend that authors test the indirect effects also with the usage of the path analysis approach. PROCESS uses OLS estimation method, which can be biased due to ignoring the measurement error. What is more, testing 12 models separately, increases the possibility of Type I errors. Using Maximum Likelihood estimation methods and testing all indirect effects in one analysis would additionally strengthen the authors’ conclusions.

Author Response

Thank you for raising this point. To follow your suggestion, we have performed a path analysis (using the Lavaan Package in R), with maximum likelihood estimation, and tested all 12 models in one analysis (again 5000 bootstraps; saturated model). Despite some small changes in the estimated coefficients, results were highly similar and indirect effects were (again) found for all associations of interest. We agree that reporting these outcomes can strengthen our conclusions, and we have therefore added the outcomes of this path-analysis to our supplement (Table S4), which we refer to in the main text (line 500 – 504). We have also included the R code of this analysis in an additional supplement (Supplement III)

“Similar effects were observed when all paths (direct and indirect, for both NEG and POS, and for all outcome variables of interest) were evaluated using a path-analysis approach in R (using the Lavaan Package, with Maximum Likelihood estimation; see Supplement II Table S4; the R code is included in Supplement III). This shows that our results are robust.”
